# Post-COVID-19 Symptoms and Heart Disease: Incidence, Prognostic Factors, Outcomes and Vaccination: Results from a Multi-Center International Prospective Registry (HOPE 2)

**DOI:** 10.3390/jcm12020706

**Published:** 2023-01-16

**Authors:** Ivan J. Núñez-Gil, Gisela Feltes, María C. Viana-Llamas, Sergio Raposeiras-Roubin, Rodolfo Romero, Emilio Alfonso-Rodríguez, Aitor Uribarri, Francesco Santoro, Víctor Becerra-Muñoz, Martino Pepe, Alex F. Castro-Mejía, Jaime Signes-Costa, Adelina Gonzalez, Francisco Marín, Javier Lopez-País, Enrico Cerrato, Olalla Vázquez-Cancela, Carolina Espejo-Paeres, Álvaro López Masjuan, Lazar Velicki, Ibrahim El-Battrawy, Harish Ramakrishna, Antonio Fernandez-Ortiz, Julián Perez-Villacastín

**Affiliations:** 1Cardiology Department, Hospital Clínico San Carlos, 28040 Madrid, Spain; 2Faculty of Biomedical and Health Sciences, Universidad Europea de Madrid, 28670 Villaviciosa de Odón, Spain; 3Cardiology Department, Hospital Vithas Arturo Soria, 28043 Madrid, Spain; 4Cardiology Department, Hospital Universitario de Guadalajara, 19002 Guadalajara, Spain; 5Cardiology Department, Hospital Universitario Álvaro Cunqueiro, 36213 Vigo, Spain; 6Emergency Department, Hospital Isabel Zendal and Hospital Universitario de Getafe, 28905 Madrid, Spain; 7Cardiology Department, Institute of Cardiology and Cardiovascular Surgery, Havana C.P. 10400, Cuba; 8Cardiology Department, Hospital Universitari Vall d’Hebron, 08035 Barcelona, Spain; 9Cardiology Department, Ospedali Riuniti, 71122 Foggia, Italy; 10Cardiology Department, Hospital Clínico Universitario Virgen de la Victoria, 29010 Málaga, Spain; 11Department of Emergency and Organ Transplantation, Cardiology Unit, University of Bari Aldo Moro, 70121 Bari, Italy; 12Cardiology Department, Hospital General del Norte de Guayaquil IESS Los Ceibos, Guayaquil 090615, Ecuador; 13Pneumology Department, Hospital Clínico de Valencia, 46010 Valencia, Spain; 14Anesthesiology Department, Hospital Universitario Infanta Sofia, 28703 San Sebastian de los Reyes, Spain; 15Cardiology Department, Hospital Clínico Universitario Virgen de la Arrixaca, 30003 Murcia, Spain; 16Cardiology Department, Complejo Hospitalario Universitario de Ourense, 32005 Orense, Spain; 17Cardiology Department, San Luigi Gonzaga University Hospital, Orbassano and Rivoli Infermi Hospital, 10043 Rivoli, Italy; 18Preventive Department, Hospital Universitario de Santiago de Compostela, 15706 Santiago de Compostela, Spain; 19Cardiology and Emergency Department, Hospital Universitario Príncipe de Asturias, 28034 Madrid, Spain; 20Cardiology Department, Hospital Universitario Juan Ramón Jimenez, 21002 Huelva, Spain; 21Faculty of Medicine, University of Novi Sad, 21137 Novi Sad, Serbia; 22Institute of Cardiovascular Diseases Vojvodina, 21204 Novi Sad, Serbia; 23Department of Cardiology and Angiology, Bergmannsheil Bochum, Medical Clinic II, Ruhr University, 44801 Bochum, Germany; 24Faculty of Medicine, University of Heidelberg, 69120 Heidelberg, Germany; 25Anesthesiology Department, Mayo Clinic, Rochester, MN 55905, USA

**Keywords:** COVID-19, mortality, cardiology, persistent, prognosis, heart disease

## Abstract

Background: Heart disease is linked to worse acute outcomes after coronavirus disease 2019 (COVID-19), although long-term outcomes and prognostic factor data are lacking. We aim to characterize the outcomes and the impact of underlying heart diseases after surviving COVID-19 hospitalization. Methods: We conducted an analysis of the prospective registry HOPE-2 (Health Outcome Predictive Evaluation for COVID-19-2, NCT04778020). We selected patients discharged alive and considered the primary end-point all-cause mortality during follow-up. As secondary main end-points, we included any readmission or any post-COVID-19 symptom. Clinical features and follow-up events are compared between those with and without cardiovascular disease. Factors with *p* < 0.05 in the univariate analysis were entered into the multivariate analysis to determine independent prognostic factors. Results: HOPE-2 closed on 31 December 2021, with 9299 patients hospitalized with COVID-19, and 1805 died during this acute phase. Finally, 7014 patients with heart disease data were included in the present analysis, from 56 centers in 8 countries. Heart disease (+) patients were older (73 vs. 58 years old), more frequently male (63 vs. 56%), had more comorbidities than their counterparts, and suffered more frequently from post-COVID-19 complications and higher mortality (OR _heart disease_: 2.63, 95% CI: 1.81–3.84). Vaccination was found to be an independent protector factor (HR _all-cause death_: 0.09; 95% CI: 0.04–0.19). Conclusions: After surviving the acute phase, patients with underlying heart disease continue to present a more complex clinical profile and worse outcomes including increased mortality. The COVID-19 vaccine could benefit survival in patients with heart disease during follow-up.

## 1. Introduction

The condition known as coronavirus 2019 disease (COVID-19) was classified as a pandemic by the World Health Organization (WHO) on 11 March 2020 and has caused millions of casualties to date [1]. In fact, as of 23 December 2022, the WHO recorded 651,918,402 confirmed cases of COVID-19, including 6,656,601 deaths worldwide [1]. As of 13 December 2022, a total of 13,008,560,983 vaccine doses had been administered also [1]. Thus, two years after the beginning of the outbreak, an increasing number of reports and studies have been published. Indexed in PubMed alone, there are 89,237 citations from 2020 and 129,416 from 2021. At the end of 2022, 47,067 were related to COVID-19 and “vaccination” as recorded in PubMed.

Several reports have evaluated the acute phase of the disease but limited long-term follow-up information is available because COVID-19 is a recently discovered condition. Interestingly, it has been linked to common cardiac complications as well [2,3,4]. Existing evidence has been reviewed and summarized in meta-analyses [2,3] and indicates frequent lasting effects after the acute phase of COVID-19 [2]. This persistent condition has been baptized by several authors with different terms such as Post-acute COVID-19, Persistent COVID-19, post-COVID syndrome, long haulers, long COVID-19, etc. The term “long COVID” was introduced by Perego on social media to signify the persistence of symptoms weeks or months after the initial COVID-19 infection, and the term ‘long haulers’ was first described by Watson et al. to identify post-COVID conditions [4]. Potential mechanisms related to the pathophysiology of long COVID-19 could include (1) harm to tissues and cells that are relevant for vascular flow, therefore blood clotting is increased; (2) persistence of fragments of the virus or its sub-particles/protein material in a wide range of tissues; and (3) an altered immune system, among others. Moreover, chronic cardiovascular effects of COVID-19 have been frequently reported, even among those who had no previous CVD, suggesting an important tropism of this virus with the cardiovascular system [5]. Therefore, heart disease patients are specifically considered to have a worse prognosis during the acute phase of severe acute respiratory syndrome coronavirus 2 (SARS-CoV-2) infection [6,7,8,9,10]. Still, specific long-term data for this cohort of patients are lacking in the literature. For this reason, we aim to characterize the outcomes and impact of these underlying heart diseases and vaccination after surviving COVID-19 hospitalization.

## 2. Methods

### 2.1. Study Design and Participation Criteria

The HOPE-2 (Health Outcome Predictive Evaluation for COVID-19-2, NCT04778020) registry is a prospective international investigator-initiated study, designed as a real-life all-comers protocol, without any financial remuneration for researchers. The study was designed as a registry. Thus, no formal sample size calculation was performed. The data that support the findings of this study are available from the corresponding author upon reasonable request. Patients were eligible for enrollment when discharged after in-hospital admission with COVID-19, dead or alive. Confirmed cases were those with positive throat swab samples tested using real-time reverse transcriptase–polymerase chain reaction assays according to the WHO recommendations. All decisions and clinical procedures were performed by the attending physician team following the local regular practice and protocols.

HOPE-2 was approved by the research ethic committee of the Hospital Clínico San Carlos (21/128-E) and was also appraised and accepted by the institutional board or local committees. Written informed consent was waived because of its anonymized observational design. The data were collected in electronic format in a secure online database [8]. The information presented here corresponds to the final cutoff performed on 31 December 2021. All local principal researchers reviewed the draft and vouched for the accuracy and veracity of the data. A complete list of participants and definitions is available in the addendum. Patients did not participate in the design or execution of the study.

#### 2.1.1. Data Collection and Variable Definitions

We assumed a pragmatic definition of previous “heart disease” according to the local research team led by 2 experienced local physicians. We considered heart disease (+) when it was stated so in the clinical history and the patient received specific medication in the follow-up for that purpose. We recognized the following categories for the main heart problems of every patient: Arrhythmias, coronary artery disease, heart failure or cardiomyopathy, heart valve disease, combined (when a combination of the former problems was present to a clinically relevant degree), and non-specified or other different from the mentioned groups (i.e., congenital heart disease). In-hospital (invasive mechanical ventilation, non-invasive mechanical ventilation, prone, respiratory insufficiency, heart failure, renal failure, upper respiratory tract involvement, pneumonia, sepsis, systemic inflammatory response syndrome, clinically relevant bleeding, hemoptysis, and embolic events) and follow-up events were allocated and gathered following local researchers’ criteria using the registry definitions. Patients were considered vaccinated when at least one dose of any brand was administered. The complete list of variables recorded in HOPE 2 is provided in the addendum.

#### 2.1.2. Study Follow-up and Outcomes

For this study, we selected patients discharged alive after COVID-19 with any previous heart condition. We considered all-cause mortality as the primary end-point. As secondary main end-points, we included any readmission and any long-term COVID-19 symptom as recorded in the clinical history or by the patient, during the clinical follow-up after discharge. A list of relevant post-COVID-19 symptoms was also recorded based on previous reports [2] (addendum). The follow-up was performed by reviewing electronic records and contact with the patient, family, or the patient’s referring physician by phone.

### 2.2. Statistical Analysis

Data are provided as mean (standard deviation) or median (interquartile range) for continuous variables, and as frequency (%) for categorical variables. Missing data were excluded (listwise deletion) and no imputation techniques were used. Student *t*-tests were used to compare continuous variables when needed. A Chi-square test was used to compare categorical variables. Univariate analysis was performed, and given the multiplicity of variables, only factors with *p* < 0.05 in the univariate analysis (heart disease cohort) were entered into the Cox multivariate regression analysis models (step-backward, Wald) to avoid overfitting in order to define independent risk factors for the principal outcome. Multicollinearity was disclosed (correlation matrix, Variance Inflation Factor (VIF), and the Tolerance Statistic values together with Eigen values, condition indexes, and variance proportions estimations) in the variables included in the multivariable model. Mortality analysis was completed using Kaplan–Meier estimates and log-rank tests to compare groups. Survival time was calculated from the admission date to the last follow-up date (or death). Statistical analysis was performed with SPSS statistics v24.0 (SPSS, Inc., Chicago, IL, USA) and Stata v14 (Statacorp, College Station, TX, USA) in the mentioned analyses. The statistical tests were 2-sided, and a *p*-value < 0.05 was considered statistically significant.

## 3. Results

### 3.1. Overall Data and General Info

HOPE-2 closed its database on 31 December 2021, with 9299 patients hospitalized with COVID-19, primarily during the first wave. Of those, 1805 patients died during the index admission and 7263 patients were discharged alive. The crude all-cause mortality odds ratio of the heart disease cohort was 2.93 (Figure 1).

For the purpose of the main objective of this study, all-cause mortality, 7014 patients were included in the present analysis, from 56 centers in 8 countries. Of those, 1163 (16.6%) were recorded to suffer from any underlying heart condition diagnosed before their index admission. The overall cohort had a mean age of 60.4 (17.4) years old, with a predominance of male gender (57.1%) and frequent cardiovascular risk factors (hypertension of 43.3%, dyslipidemia of 29.0%, diabetes of 15.4%, obesity of 17.5%, and smoking habit of 6.0%).

### 3.2. Heart Disease (+) vs. Heart Disease (−) Cohorts

Heart disease (+) patients were older (mean, 73 vs. 58 years old), more frequently male (63 vs. 56%), and had more cardiovascular risk factors and comorbidities than their counterparts without heart disease (−) (Table 1).

During admission, heart disease (+) patients suffered more frequently from complications, needing more commonly advanced circulatory support and presented more unfavorable features at discharge. Despite this different risk profile, we found no differences in the rate of vaccination among the survivors with heart disease or not (53.7 vs. 51.4%, respectively, *p* = 0.371).

The crude mortality rate after a mean follow-up of 3.8 (5.4) months was still higher in the heart disease (+) cohort (OR_all-cause death_: 2.63; 95% CI: 1.81–3.84, *p* < 0.001).

Death and readmission rates were, overall, higher in the heart disease (+) group (Figure 2). However, considering only vaccinated patients, the mortality differences between cohorts disappeared.

### 3.3. Heart Disease (+) Patients and Mortality

Assessing only all-cause mortality in the heart disease (+) cohort, we found that deceased patients, after a mean follow-up of 4.3 (5.7) months, were older, had worse renal function, and more oncologic comorbidities (Table 2).

During COVID-19 admission, those who died during follow-up suffered more events and, at discharge, displayed higher levels of d-dimer, PCR, and NT-proBNP. They also faced more readmissions, as shown in Table 2. Noteworthily, vaccinated patients had a lower risk of mortality (1.9% vs. 5.7%; *p* < 0.001; univariate OR: 0.32, 95% CI: 0.20–0.50) and of readmission (16.2% vs. 19.2%, univariate *p* = 0.04; OR: 0.81; 95% CI: 0.67–0.99). However, no clear differences in survival were found among subtypes of heart disease.

In the multivariate analysis for all-cause mortality, we included age (>70 years), renal failure, any cancer, any dependency level, and vaccination status in the model. Age (>70 years old; HR_all-cause death_: 6.93; 95% CI: 2.27–21.12; *p* = 0.001), any level of dependence (HR_all-cause death_: 4.38, 95% CI: 2.19–8.74; *p* < 0.001), and COVID-19 vaccination (HR_all-cause death_: 0.09; 95% CI: 0.04–0.19; *p* < 0.001) remained in the model as independent predictors of the outcome.

### 3.4. Symptoms Post-COVID-19 and Vaccination

Finally, 2651 records were available for specific data on long-term post-COVID-19 symptoms. After discharge, we frequently found post-COVID-19 symptoms (Figure 3). Most of these symptoms were more common in the heart disease (+) group. Considering only those with clinical events after discharge, the mean time to complete symptom recovery was significantly different between the two groups (3.33 vs. 3.93 months; Welch-t test, *p* = 0.048). Of 1692 patients with NYHA I before the COVID-19 admission, 145 worsened after discharge (129 NYHA II; 14 NYHA III and 2 NYHA IV).

In relation to the vaccine, vaccinated patients in the heart disease (+) cohort displayed fewer symptoms during follow-up. Considering the list of symptoms displayed in Figure 3, we found the following risk estimations for vaccinated heart disease (+) patients with statistical significance: Fatigue (OR:0.58, 95% CI 0.39–0.86, *p* = 0.006), dizziness (OR:0.40, 95% CI 0.23–0.69, *p* = 0.001), chest pain (OR:0.48, 95% CI 0.26–0.88, *p* = 0.016), acute coronary syndromes (OR:0.12, 95% CI 0.03–0.53, *p* = 0.001), palpitations (OR:0.33, 95% CI 0.20–0.55, *p* < 0.001), resting heart rate increase (OR:0.31, 95% CI 0.15–0.66, *p* = 0.001), any arrythmia (OR:0.34, 95% CI 0.20–0.56, *p* < 0.001), new LV dysfunction (OR:0.26, 95% CI 0.10–0.73, *p* = 0.006), migraine (OR:0.06 95% CI 0.01–0.49, *p* < 0.001), anxiety (OR:0.44, 95% CI 0.26–0.76, *p* = 0.003), depression (OR:0.45, 95% CI 0.24–0.84, *p* = 0.010), mood disorder (OR:0.28, 95% CI 0.14–0.57, *p* < 0.001), tongue involvement (OR:0.48, 95% CI 0.44–0.53, *p* = 0.022), and chills (OR:0.14, 95% CI 0.03–0.63, *p* = 0.003).

Additionally, we found a favorable trend (all *p* < 0.10) in vaccinated patients regarding myopericarditis (3 vs. 0 cases, *p* = 0.07), atrial fibrillation, and tinnitus/hearing loss. The only symptom detected more frequently in vaccinated patients was weight loss (OR: 5.05, 95% CI 1.89–13.50, *p* < 0.001).

## 4. Discussion

Herein, we assess the impact of underlying heart diseases on prognosis after hospitalization with COVID-19, the clinical predictors, and the importance of vaccination in a large prospective multi-center international registry on COVID-19. Our main findings are the following. Heart disease (+) patients hospitalized with COVID-19 were older, with more comorbidities and higher mortality during a hospital stay. Patients with heart disease presented higher mortality rates and more complications during follow-up after COVID-19 discharge. Interestingly, COVID-19 vaccination depicted an important beneficial influence on mortality in the whole cohort. In fact, differences in evolutive mortality in vaccinated heart disease (+) patients compared with the heart disease (−) group grossly disappeared. Furthermore, post-COVID-19 complaints were more frequent in the heart disease (+) cohort. Fatigue was the most common symptom found after COVID-19 in heart disease (+) and (−) patients. Furthermore, our study suggests that vaccination could decrease several symptoms in the subacute phase in patients with heart disease.

It is known that patients with cardiovascular risk factors (i.e., hypertension, renin angiotensin aldosterone system abnormalities, etc.) or underlying cardiovascular disease have worse outcomes [8]. Moreover, patients with COVID-19 have been reported to also develop cardiovascular and thrombotic complications in the absence of underlying cardiovascular disease. Our data show both circumstances. In this setting, some cardiovascular drugs such ACEIs/ARBs or aspirin, after wide scientific discussion, have demonstrated potential benefits [11,12,13,14,15]. We should ensure that patients who are suffering from heart disease receive these drugs when indicated in order to improve the function of their renin-angiotensin-aldosterone system or thrombogenic status.

One of the most potentially relevant findings of this study is the clear beneficial effect of COVID-19 vaccines in the heart disease (+) cohort. Recently, some COVID-19 vaccines raised certain concerns [16], primarily on social media rather than in the scientific community, about cardiovascular effects such as myocarditis [11,17]. However, this risk after millions of doses is very small and usually leads to mild cardiac inflammations with fast recovery and no short-term complications [17]. Additionally, some researchers have pointed out their concerns about the possible association between mRNA vaccines and the increased risk of acute coronary syndromes [11,18]. The latter, an observational report, included 566 patients and measured only surrogate points (i.e., IL-16, Fas, and HGF levels) without a control group. In contrast, the results of our study suggest a potential effect on the survival and symptomatic benefit of COVID-19 vaccines in a clear setting of fragility for the infection. In addition, the rates of peri/myocarditis and acute coronary syndromes in our vaccinated patients with heart disease were lower than in unvaccinated patients with heart disease, among other symptomatic benefits of vaccination.

Lopez-Leon et al. [2], in a recent meta-analysis including 15 papers and 47,910 patients, identified more than 50 post-COVID-19 effects, ranging from 14 to 110 days post-infection. According to this study, the five most common manifestations were fatigue (58%, 95% CI 42–73), headache (44%, 95% CI 13–78), attention disorder (27%, 95% CI 19–36), hair loss (25%, 95% CI 17–34), and dyspnea (24%, 95% CI 14–36). In the same line, fatigue was also the most frequent symptom in our heart disease (+) patients (44.8%) and the control group (24.3%). Despite the high heterogeneity of the data sources from all those studies, it is clear that COVID-19 poses a high risk of postinfectious symptoms in the short term. Symptoms observed in post-COVID-19 patients could resemble the chronic fatigue syndrome, which has been deemed to include severe disabling fatigue, pain, neurocognitive impairment, sleep disorders, and worsening of global symptoms following minor increases in physical and/or cognitive activity. This syndrome has also been linked to other viruses (Epstein–Barr, cytomegalovirus, enterovirus, and herpesvirus), immune dysfunction, endocrine-metabolic dysfunction, and neuropsychiatric factors [2].

Recently, the WHO developed a Delphi consensus regarding the clinical definition of the post-COVID-19 condition [19]. The WHO document agrees with the fact that most post-COVID-19 cases occur in individuals with a history of probable or confirmed SARS-CoV-2 infection, usually 3 months after the onset of COVID-19, with symptoms that last for at least 2 months and cannot be explained by an alternative diagnosis. Our study did not completely fulfill this definition since it was designed before the publication of the consensus [19]. However, in this setting, the present study provides prospective data in order to better evaluate the natural course of COVID-19 infection and define post-COVID-19 outcomes in heart disease patients.

Likewise, a more severe presentation could be related to more evolutive symptoms as suggested in 1969 patients by the LONG-COVID-EXP-CM Multicenter Study [20]. Interestingly, symptoms may be new after the initial recovery from the acute COVID-19 episode or persist from it [19]. Moreover, symptoms may also fluctuate or relapse over time [19].

### Clinical Implications

Our results indicate the clinical relevance of the post-COVID-19 phase. From a clinical point of view, as mentioned by Lopez-Leon et al. [2], physicians should be aware of the symptoms, signs, and biomarkers present in their patients previously affected by COVID-19 to promptly assess and identify the long-COVID-19 condition. Management of all these effects likely needs further understanding to design individualized, dynamic cross-sectoral protocols in “Post-COVID-19” clinics [21] with multiple specialties, including rehabilitation with graded exercise, physical therapy, frequent medical evaluations, and cognitive behavioral therapy when required.

In this complex setting, the implementation of new arising techniques such as individual genetics or state-of-the-art disease markers such as microRNAs could lead to the identification of molecular patterns useful in early diagnosis and outcome predictions, potentially leading to improvements in management and perhaps in the natural history of the disease [22].

The vaccination arises as a very useful tool to potentially improve the prognosis of patients with heart disease. In this regard, only future studies with extensive long-term data will shed light on the complete natural history of the SARS CoV-2 infection.

## 5. Limitations

The main limitation is secondary to the observational design. In addition, the specific definition of the variables [19], the type and degree of heart disease, and the reporting of the events could vary among centers, countries, and the precise moment in their pandemic curve or the predominant viral strain. In fact, we have no data available on the precise type of COVID-19 mutant strain. Moreover, the definition of “heart disease” is broad and should also be cautiously considered. Moreover, for overlapping heart conditions, we were not able to provide the relative weight of the different heart diseases in the patient’s outcomes. Some subjective issues, such as some symptoms with potential variable reporting, and their resolution should also be viewed with caution. Regarding the treatment applied or the vaccination schedules, the treating physician decided it at all times following local or national protocols. Due to differences among wave timings, virus strains, national health systems, resources, and population features, some differences between countries could exist.

Unfortunately, also due to the sample size, we could not distinguish between different types of vaccines, and our findings should be carefully considered as hypothesis generating. Adverse events were not centrally adjudicated, which could imply a certain degree of underestimation or underreporting. All this results in a heterogeneous sample, which does not produce information as robust as a clinical trial would do, and should be, again, considered hypothesis generating. Furthermore, it would be interesting to note whether the mortality risk is due to the heart disease itself or COVID-19. However, with the data available and the present design, we cannot further disclose this point.

On the other hand, these observations provide an overall real-life idea in this setting in a broad international sample with a considerable prospective follow-up.

## 6. Conclusions

Patients with underlying heart disease hospitalized with COVID-19 were older and had more comorbidities than the overall population admitted with the infection. After the acute phase, when they had more complications and higher mortality, those patients with underlying heart issues tend to have a more complex clinical profile and worse outcomes, including increased mortality. The COVID-19 vaccine could provide a clinical benefit in patients with heart disease. Our results encourage COVID-19 vaccination in patients with underlying heart diseases after COVID-19 infection.

## Figures and Tables

**Figure 1 jcm-12-00706-f001:**
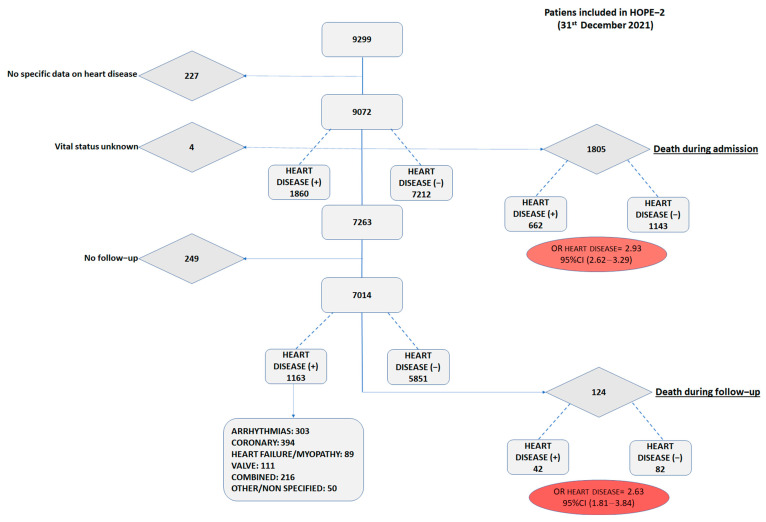
Study flow diagram.

**Figure 2 jcm-12-00706-f002:**
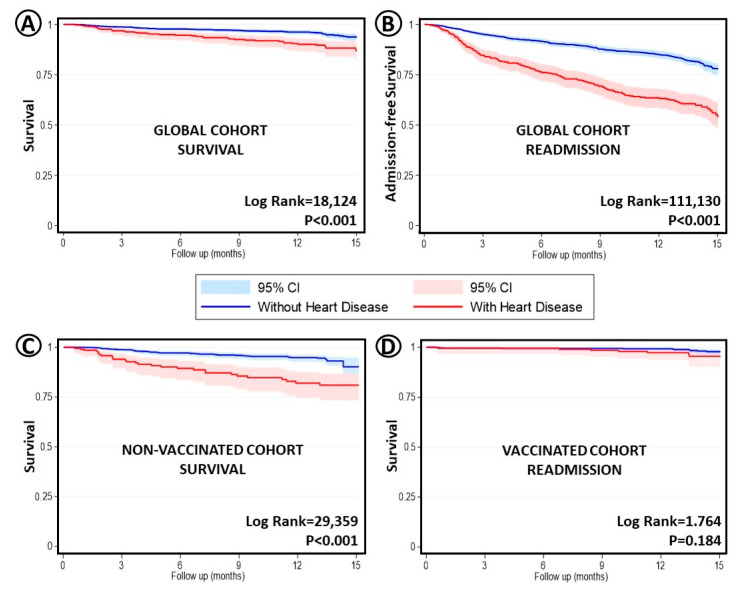
Kaplan–Meier curves unadjusted survival analysis, comparing no heart disease vs. any type of underlying heart condition. T_0_ = Admission date. (**A**) Global cohort and survival. (**B**) Same comparison but regarding readmissions. (**C**) Cohort of patients not vaccinated (any dose) and survival. (**D**) Cohort of patients vaccinated (any dose) and survival.

**Figure 3 jcm-12-00706-f003:**
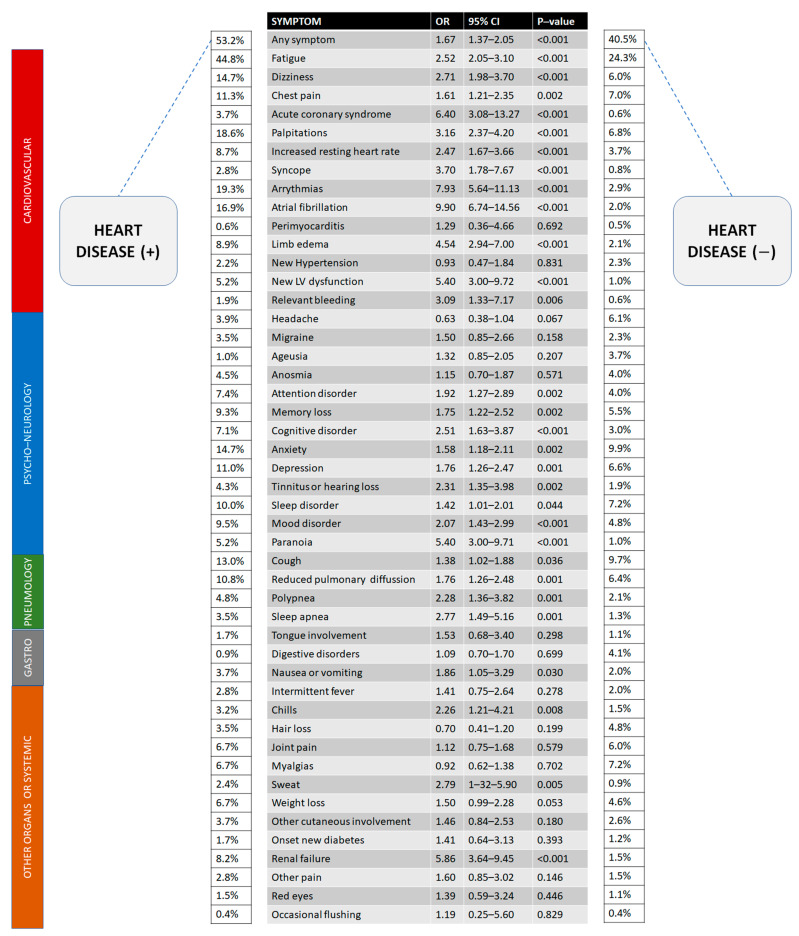
Long-term effects of COVID-19 (post-COVID-19 syndrome) after discharge comparing patients with a heart condition or not (data expressed in percentages over available patients, *N* = 2651). Risk estimates (OR) are univariate.

**Table 1 jcm-12-00706-t001:** Clinical features before admission, discharge management, vaccination, and vital status stratified by heart disease. Data presented over available registries in each variable (percentages). All these patients were discharged alive. Univariate risk estimators (odds ratio (OR) and the corresponding 95% confidence interval) are provided. * Comparing age <70 vs. ≥70 years old. ** Caucasian vs. no Caucasian. *** Any chest X-ray abnormality (No/Yes).

Characteristics /7014	Heart Disease (−) /5851	Heart Disease (+) /1163	*p*-Value	Odds Ratio (95% CI)
**BEFORE ADMISSION**				
**Gender, male**	3272 (55.9)	730 (62.8)	**<0.001**	**1.33 (1.17–1.51)**
**Age, mean (SD)**	57.9 (17.0)	72.7 (13.9)	**<0.001**	**4.98 (4.35–5.70) ***
**Race** - **Black** - **Caucasian** - **Latin** - **Asiatic** - **Other**	73 (1.2) 4389 (75.0) 978 (16.7) 362 (6.2) 49 (0.8)	26 (2.2) 1005 (86.4) 110 (9.5) 14 (1.2) 8 (0.7)	**<0.001**	**0.47** **(0.** **39–0.56) ****
**Health care professional, no**	4989 (85.3)	1084 (93.2)	**<0.001**	**0.42 (0.33–0.54)**
**Hypertension**	2152 (36.8)	885 (76.1)	**<0.001**	**5.47 (4.73–6.32)**
**Dyslipidemia**	1421 (24.3)	611 (52.5)	**<0.001**	**3.45 (3.03–3.93)**
**Diabetes Mellitus**	761 (13.3)	322 (28.5)	**<0.001**	**2.60 (2.24–3.02)**
**Obesity**	960 (16.4)	264 (22.7)	**<0.001**	**1.50 (1.28–1.74)**
**Smoker**	334 (6.3)	88 (8.3)	**0.018**	**1.34 (1.05–1.72)**
**Renal insufficiency**	165 (2.8)	124 (10.7)	**<0.001**	**4.11 (3.23–5.24)**
**Any lung disease**	770 (16.9)	323 (35.0)	**<0.001**	**2.64 (2.26–3.10)**
**Any cerebrovascular disease**	225 (3.8)	153 (13.2)	**<0.001**	**3.79 (3.05–4.70)**
**Connective disease**	120 (2.1)	41 (3.5)	**0.002**	**1.74 (1.22–2.50)**
**Liver disease**	153 (2.6)	58 (5.0)	**<0.001**	**1.95 (1.43–2.66)**
**Any cancer**	525 (9.0)	161 (13.8)	**<0.001**	**1.63 (1.35–1.97)**
**Any immunosuppression**	307 (5.2)	90 (7.7)	**0.001**	**1.51 (1.19–1.93)**
**Any dependency level**	352 (6.3)	206 (18.2)	**<0.001**	**3.32 (2.76–4.00)**
**Home oxygen therapy**	74 (1.3)	56 (4.8)	**<0.001**	**3.94 (2.77–5.62)**
**Oral anticoagulation therapy**	106 (1.8)	416 (35.8)	**<0.001**	**30.18 (24.07–37.85)**
**ACEI/ARB therapy**	1484 (25.4)	658 (56.6)	**<0.001**	**3.83 (3.37–4.37)**
**DURING ADMISSION**				
**Hemoglobin at admission (SD)**	13.71 (1.71)	13.07 (2.07)	**<0.001**	**-**
**Creatinine at admission (SD)**	0.98 (0.88)	1.34 (1.21)	**<0.001**	**-**
**Elevated D dimer, at admission**	2925 (50.1)	628 (54.1)	**0.013**	**1.17 (1.03–1.33)**
**Elevated procalcitonin, at admission**	630 (10.8)	181 (15.6)	**<0.001**	**1.53 (1.28–1.82)**
**Elevated PCR, at admission**	4563 (78.2)	993 (85.5)	**<0.001**	**1.65 (1.38–1.97)**
**Elevated Troponin, at admission**	284 (4.9)	186 (16.0)	**<0.001**	**3.73 (3.06–4.54)**
**Abnormal chest X ray, at admission** - **Unilateral findings** - **Bilateral**	1055 (20.1) 3313 (63.0)	230 (22.1) 599 (57.6)	**0.003**	**0.80 (.058–0.94) *****
**Abnormal blood pressure**	242 (4.6)	88 (8.5)	**<0.001**	**1.92 (1.49–2.47)**
**Heart failure**	107 (1.8)	166 (14.4)	**<0.001**	**8.47 (6.97–11.55)**
**Renal failure**	409 (7.0)	215 (18.6)	**<0.001**	**3.03 (2.53–3.62)**
**Sepsis**	269 (4.6)	82 (7.1)	**<0.001**	**1.58 (1.22–2.04)**
**SIRS**	800 (13.7)	173 (15.0)	**0.264**	**1.11(0.93–1.32)**
**Relevant bleeding**	83 (1.4)	49 (4.2)	**<0.001**	**3.07 (2.14–4.39)**
**Hemoptysis**	60 (1.0)	24 (2.1)	**0.003**	**2.04 (1.26–3.29)**
**Embolic event**	99 (1.7)	28 (2.4)	**0.092**	**1.44 (0.94–2.20)**
**NIMV**	589 (10.1)	158 (13.7)	**<0.001**	**1.41 (1.17–1.70)**
**IMV**	265 (4.6)	67 (5.8)	**0.067**	**1.30 (0.98–1.70)**
**ECMO or other circulatory support**	21 (0.4)	9 (0.9)	**0.043**	**2.12 (1.54–2.91)**
**AFTER DISCHARGE**				
**Discharge anticoagulation**	1321 (22.7)	562 (48.8)	**<0.001**	**9.44 (7.98–11.17)**
**Abnormal LVEF (<54%) at discharge**	21 (2.4)	58 (22.5)	**<0.001**	**11.90 (7.05–20.46)**
**Elevated D dimer, at discharge**	599 (27.5)	152 (33.0)	**0.017**	**1.30 (1.05–1.61)**
**Elevated procalcitonin, at discharge**	137 (6.3)	62 (13.4)	**<0.001**	**2.32 (1.69–3.19)**
**Elevated PCR, at discharge**	741 (34.0)	206 (44.7)	**<0.001**	**1.57 (1.28–1.92)**
**Elevated Troponin, at discharge**	48 (2.2)	33 (7.2)	**<0.001**	**3.43 (2.17–5.40)**
**Elevated NTproBNP, at discharge**	50(2.3)	55 (11.9)	**<0.001**	**5.77 (3.88–8.59)**
**Abnormal chest X ray, at discharge**	412 (27.7)	123 (37.5)	**<0.001**	**1.56 (1.22–2.01)**
**Abnormal spirometry, at follow-up**	67 (7.8)	33 (18.8)	**<0.001**	**2.72 (1.73–4.29)**
**Abnormal chest CT scan, at follow-up**	183 (22.9)	59 (33.7)	**0.003**	**1.71 (1.20–2.47)**
**Readmission during follow-up**	261 (11.8)	158 (33.5)	**<0.001**	**3.77 (3.00–4.75)**
**New COVID-19 episode**	58 (2.6)	11 (2.3)	**0.722**	**0.89 (0.46–1.71)**
**Any clinical event after discharge**	886 (40.5)	246 (53.2)	**<0.001**	**1.67 (1.37–2.05)**
**Vaccination**	1138 (51.4)	253 (53.7)	**0.371**	**1.09 (0.90–1.34)**
**Vaccine brand** - **Astra Zeneca** - **Johnson and Johnson/Janssen** - **Moderna** - **Pfizer** - **Sinovac** - **Others** - **Unknown**	157 (2.7) 36 (0.6) 98 (1.7) 831 (14.2) 0 6 (0.1) 10 (0.2)	20 (1.7) 6 (0.5) 29 (2.5) 192 (16.5) 2 (0.2) 1 (0.1) 3 (0.3)	**0.001**	
**Death during follow-up**	82 (1.4)	42 (3.6)	**<0.001**	**2.64 (1.81–3.85)**

**Table 2 jcm-12-00706-t002:** Clinical features of patients with any underlying heart disease before admission, discharge management, and vaccination stratified by vital status after a mean follow-up. Data presented over available registries in each variable (percentages). All these patients were discharged alive. *p* < 0.05 variables, before admission, were included in the multivariate model (see text). Univariate risk estimators (odds ratio (OR) and the corresponding 95% confidence interval) are provided. * Comparing age <70 vs. ≥70 years old. ** Caucasian vs. no Caucasian. *** Any chest X-ray abnormality (No/Yes).

Characteristics of Heart Disease Patients /1163	Alive /1121	Deceased /42	*p*-Value **	Odds Ratio (95% CI)
**BEFORE ADMISSION**				
**Gender, male**	704 (62.8)	26 (61.9)	0.906	0.96 (0.51–1.81)
**Age, mean (SD)**	72.3 (13.8)	83.4 (9.9)	**0.005**	**4.23 (1.65–10.84) ***
**Race** - **Black** - **Caucasian** - **Latin** - **Asiatic** - **Other**	26 (2.3) 963 (85.9) 110 (9.8) 14 (1.2) 8 (0.7)	0 42 (100) 0 0 0	0.144	0.96 (0.95–0.97) **
**Health care professional, no**	1044 (93.1)	40 (95.2)	0.934	0.68 (0.16–2.59)
**Type of (main) heart disease** - **Arrhythmias** - **Combined** - **Coronary** - **Heart failure/myopathy** - **Not disclosed** - **Valve**	294 (26.2) 203 (18.1) 384 (34.3) 85 (7.6) 48 (4.3) 107 (9.5)	9 (21.4) 13 (31.0) 10 (23.8) 4 (9.5) 2 (4.8) 4 (9.5)	0.360	-
**Hypertension**	850 (75.8)	35 (83.3)	0.263	1.59 (0.70–3.63)
**Dyslipidemia**	589 (52.5)	22 (52.4)	0.984	0.99 (0.54–1.84)
**Diabetes Mellitus**	312 (28.7)	10 (25.0)	0.616	0.83 (0.40–1.72)
**Obesity**	255 (22.7)	9 (21.4)	0.841	0.93 (0.44–1.96)
**Smoker**	86 (8.4)	2 (4.9)	0.419	0.56 (0.13–2.35)
**Renal insufficiency**	114 (10.2)	10 (23.8)	**0.005**	**2.76 (1.32–5.76)**
**Any lung disease**	305 (34.5)	18 (48.6)	0.076	1.80 (0.93–3.48)
**Any cerebrovascular disease**	145 (12.9)	8 (19.0)	0.250	1.58 (0.72–3.49)
**Connective disease**	38 (3.4)	3 (7.1)	0.195	2.20 (0.65–7.41)
**Liver disease**	55 (4.9)	3 (7.1)	0.513	1.49 (0.45–4.98)
**Any cancer**	148 (13.2)	13 (31.0)	**0.001**	**2.95 (1.50–5.80)**
**Any immunosuppression**	84 (7.5)	6 (14.3)	0.106	2.06 (0.84–5.02)
**Any dependency level**	186 (17.1)	20 (48.8)	**<0.001**	**4.62 (2.45–8.69)**
**Home oxygen therapy**	54 (4.8)	2 (4.8)	0.987	0.99 (0.23–4.20)
**Oral anticoagulation therapy**	394 (35.1)	22 (52.4)	0.022	2.03 (1.10–3.76)
**ACEI/ARB therapy**	638 (56.9)	20 (47.6)	0.233	0.69 (0.87–1.28)
**DURING ADMISSION**				
**Elevated D dimer, at admission**	601 (53.7)	27 (64.3)	0.177	1.55 (0.81–2.95)
**Elevated procalcitonin, at admission**	175 (15.6)	6 (14.3)	0.812	0.90 (0.37–2.17)
**Elevated PCR, at admission**	954 (85.3)	39 (92.9)	0.169	2.25 (0.69–7.36)
**Elevated Troponin, at admission**	176 (15.7)	10 (23.8)	0.161	1.67 (0.81–3.47)
**Abnormal chest X ray, at admission** - **Unilateral findings** - **Bilateral**	218 (21.8) 579 (57.8)	12 (30.8) 20 (51.3)	0.414	1.17 (0.51–2.69) ***
**Abnormal blood pressure**	80 (8.0)	8 (22.9)	**0.002**	**3.43 (1.51–7.80)**
**Heart failure**	155 (13.9)	11 (26.2)	**0.026**	**2.20 (1.08–4.45)**
**Renal failure**	199 (17.9)	16(1.4)	**0.001**	**2.82 (1.49–5.36)**
**Sepsis**	75 (6.7)	7 (16.7)	0.014	2.76 (1.19–6.43)
**SIRS**	164 (14.7)	9 (21.4)	0.234	1.58 (0.74–3.36)
**Relevant bleeding**	45 (4.0)	4 (9.5)	0.084	2.50 (0.85–7.29)
**Hemoptysis**	24 (2.2)	0	0.336	-
**Embolic event**	27 (2.4)	1 (2.4)	0.984	0.98 (0.13–7.38)
**NIMV**	155 (13.9)	3 (7.1)	0.209	0.47 (0.14–1.56)
**IMV**	66 (6.0)	1 (2.4)	0.332	038 (0.05–2.84)
**ECMO or other circulatory support**	9 (0.9)	0	0.529	0.96 (0.94–0.97)
**AFTER DISCHARGE**				
**Discharge anticoagulation**	171 (39.4)	8 (27.6)	0.206	2.39 (0.65–8.79)
**Abnormal LVEF (<54%) at discharge**	54(21.8)	4 (40.0)	0.176	2.39 (0.65–8.79)
**Elevated D dimer, at discharge**	136 (31.5)	16 (55.2)	**0.009**	**2.68 (1.25–5.72)**
**Elevated procalcitonin, at discharge**	57 (13.2)	5 (17.2)	0.536	1.37 (0.50–3.74)
**Elevated PCR, at discharge**	185 (42.8)	21 (72.4)	**0.002**	**3.50 (1.52–8.09)**
**Elevated Troponin, at discharge**	29 (6.7)	4 (13.8)	0.152	2.22 (0.72–6.82)
**Elevated NTproBNP, at discharge**	47 (10.9)	8 (27.6)	**0.007**	**3.12 (1.31–7.44)**
**Abnormal chest X ray, at discharge**	115 (36.7)	8 (53.3)	0.195	1.97 (0.69–5.57)
**Abnormal spirometry, at follow-up**	33 (19.2)	0	0.331	-
**Abnormal chest CT scan, at follow-up**	54 (32.1)	5 (71.4)	**0.031**	**5.28 (0.99–28.08)**
**Readmission during follow-up**	135 (31.1)	23 (62.2)	**<0.001**	**3.64 (1.82–7.29)**
**New COVID-19 episode**	11 (2.5)	0	0.327	0.92 (0.89–0.94)
**Any clinical event after discharge**	229 (52.9)	17 (58.6)	0.549	1.62 (0.59–2.71)
**Vaccination**	245 (56.5)	8 (21.6)	**<0.001**	**0.21 (0.9–0.47)**
**Vaccine brand** - **Astra Zeneca** - **Johnson and Johnson/Janssen** - **Moderna** - **Pfizer** - **Sinovac** - **Others** - **Unknown**	20 (1.8) 6 (0.5) 28 (2.5) 185 (16.5) 2 (0.2) 1 (0.1) 3 (0.3)	0 0 1 (2.4) 7 (16.7) 0 0 0	0.990	-

## Data Availability

Upon reasonable request to the correspondence author.

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
