# Peer review of "Post-COVID-19 Symptoms and Heart Disease: Incidence, Prognostic Factors, Outcomes and Vaccination: Results from a Multi-Center International Prospective Registry (HOPE 2)"

_jcm, 2023, doi:10.3390/jcm12020706_

Round 1

Reviewer 1 Report

The article reflects a cruel, double-reality related to the covid-19 pandemic. The first is related to the exclusion from hospital care of non-covid patients, regardless of the associated pathology and the second one is related to the immediate and remote side-effects of covid vaccination. This paper highlights the sad situation, that more elderly people with multiple comorbidities were cared for, at the expense of young cardiac adults, whose subsequent evolution is difficult to predict.

The article in its current form, is good, but with the recommendation, that in the light of the latest studies regarding the side effects of vaccination, to encourage it without reserve is not advisable. Thus, I suggest that the authors could think about a rewrite of the last sentence in their conclusions. Also, I think maybe it will be useful to add one or two more other conclusions related to the results of their observations on the evolution of the two covid-19 patient categories: with and without cardiological comorbidities.

Author Response

We thank you for taking the time to review our manuscript. We made our best to address each of the comments to improve our paper.  Here, we will answer in red between lines for readability purposes and in red in the manuscript as well.

The article reflects a cruel, double-reality related to the covid-19 pandemic. The first is related to the exclusion from hospital care of non-covid patients, regardless of the associated pathology and the second one is related to the immediate and remote side-effects of covid vaccination. This paper highlights the sad situation, that more elderly people with multiple comorbidities were cared for, at the expense of young cardiac adults, whose subsequent evolution is difficult to predict.

Absolutely.

The article in its current form, is good, but with the recommendation, that in the light of the latest studies regarding the side effects of vaccination, to encourage it without reserve is not advisable. Thus, I suggest that the authors could think about a rewrite of the last sentence in their conclusions. Also, I think maybe it will be useful to add one or two more other conclusions related to the results of their observations on the evolution of the two covid-19 patient categories: with and without cardiological comorbidities.

Ok, done. We tone down a little bit the phrase.

“The COVID-19 vaccine could provide a clinical benefit in patients with heart disease.”

Reviewer 2 Report

Overall, this work is good; it is important and leads to the positive conclusions stated by the authors. I really appreciate your work

-The introduction is inadequate; please include information regarding the many varieties of COVID-19 vaccines.
-Please, indicate the type of COVID-19 mutant strain in the target population
-Sampling technique and selection of subjects are not explained, please indicate it with reporting the SZ equation
-In method section, Different types of vaccines should be taken in more than one dose, please, evaluate the difference in effects between the first and the second doses.

-Presentation of the results should be more concise and logical,
the figures and tables should be placed within the text not in a separate part .All subtitles in result section should be numbered
 -When citing figures and tables, ‘as shown in’, ‘as presented in’, etc. or brackets were usually used other than just citing the figures as ‘Figure 2
-The quality of the Figure 1 is poor

-
What about the side effects of the different types of vaccines…how was the follow up after vaccination especially, the fact that myocarditis has been reported following many vaccines.

A more detailed conclusion is required-

Minor comments

The authors should carefully correct spelling and any typo errors in entire manuscript

It is need to make additional adjustments to the article's title

I suggest the authors to add the phrases of 'background', 'methods', 'results' and 'conclusion' in Abstract section

Results word is repeated, one with numbering and other without, Please remove the numbering and add this number to the title "Overall data and general info''

line 63: It is necessary to paraphrase it
line 75: the reference 5 should be in brackets
line 300: Add reference

Author Response

We thank you for taking the time to review our manuscript. We made our best to address each of the comments to improve our paper.  Here, we will answer in red between lines for readability purposes and in red in the manuscript also.

Overall, this work is good; it is important and leads to the positive conclusions stated by the authors. I really appreciate your work

-The introduction is inadequate; please include information regarding the many varieties of COVID-19 vaccines

Ok, we update it. We also add a small remark on vaccines. However, vaccination is just a secondary issue in the paper.

-Please, indicate the type of COVID-19 mutant strain in the target population

Unfortunately, we do not have this specific data. Thus, we include that in the limitations area. However, we provide the inclusion interval which orientates the reader regarding the dominant strain at that time. 

-Sampling technique and selection of subjects are not explained, please indicate it with reporting the SZ equation

The study was designed as a registry. Thus, no formal SZ calculation was performed. We include this statement in the Methods area.

-In method section, Different types of vaccines should be taken in more than one dose, please, evaluate the difference in effects between the first and the second doses.

Unfortunately, since the study was not designed for that purpose, we do not have these data. In fact, for the analysis purposes, patients were considered vaccinated when at least one dose of any brand was administered. 

On the other side, we have indeed the type of vaccine administered. We depict that data in the table 1. However, we did not provide statistical analyses regarding the vaccine brand because the low number preclude from getting relevant results among groups. 

-Presentation of the results should be more concise and logical, the figures and tables should be placed within the text not in a separate part .All subtitles in result section should be numbered

Right, thanks for the comment. Done.

 -When citing figures and tables, ‘as shown in’, ‘as presented in’, etc. or brackets were usually used other than just citing the figures as ‘Figure 2

Ok, we modify the manuscript.

-The quality of the Figure 1 is poor

Thanks for the warning. We produced a higher quality file (Png format, apart from the manuscript for production purposes).

-What about the side effects of the different types of vaccines…how was the follow up after vaccination especially, the fact that myocarditis has been reported following many vaccines.

That is a great question. We dealt with that in an aggregate and secondary way. This is because the registry was orientated to the disease not so to the vaccination. In addition, we had a very heterogeneous vaccination strategies setting depending the different countries. We provide some small insights thought. 

A more detailed conclusion is required-

We modified the conclusion. Reviewer 1 asked us to be more prudent regarding the vaccination issue.

Minor comments

The authors should carefully correct spelling and any typo errors in entire manuscript

Ok, thanks.

It is need to make additional adjustments to the article's title

We modify it, hope it fits better now.

Post-COVID 19 symptoms and heart disease: Incidence, prognostic factors, outcomes and vaccination. Results from a multi-center international prospective registry (HOPE 2).”

I suggest the authors to add the phrases of 'background', 'methods', 'results' and 'conclusion' in Abstract section

Done.

Results word is repeated, one with numbering and other without, Please remove the numbering and add this number to the title "Overall data and general info''

line 63: It is necessary to paraphrase it
line 75: the reference 5 should be in brackets
line 300: Add reference

 Thank you so much for the tips. We also removed the line-numbering.

Reviewer 3 Report

The manuscript by Iván J. Núñez-Gil et al. entitled “Post-COVID 19 symptoms, vaccination and heart disease. Results from HOPE 2” aimed to characterize the outcomes and the impact of underlying heart diseases after surviving a COVID19 hospitalization.

I read with great interest this paper. The abstract summarizes the general significance of the manuscript; however, the title is not catchy and it does not sum up the meaning of the article.

Moreover, some major issues need to be addressed to improve the significance of the manuscript:

- In figure 3 the image of man and woman are not suitable to indicate the presence or absence of heart disease. Please, replace them.

- It would be interesting to know the main basal blood tests of the two populations (for example hemoglobin or creatinine)

- Cardiovascular (CV) involvement is a crucial complication in COVID-19, and no strategies are available to prevent or specifically address CV events in COVID patients. The identification of molecular partners contributing to CV manifestations in COVID-19 patients is crucial for providing early biomarkers, prognostic predictors and new therapeutic targets. Specifically, miRNAs have been proposed as valuable biomarkers and predictors of both cardiac and vascular damage occurring in SARS-CoV-2 infection. Consequently, this article could be discussed: Izzo C, Visco V, Gambardella J, et al. Cardiovascular implications of miRNAs in COVID-19 [published online ahead of print, 2022 Jul 2]. J Pharmacol Exp Ther. 2022;JPET-MR-2022-001210. doi:10.1124/jpet.122.001210.

-As reported in limitations, the authors are not able to provide the relative weigh of the different heart disease in the patient’s outcome. Accordingly, it would be important to understand whether in “heart disease(+)”patients  due to an arrhythmia, the arrhythmia was still ongoing at the time of enrollment or was in the past. Similarly, did “heart disease(+)” patients due to valve disease have moderate/severe or only mild valve disease?

- Some numbers of the bibliography are reported in the text without square brackets. Please check them.

- Line 330: there is a typo “About Regarding”.

Author Response

We thank you for taking the time to review our manuscript. We made our best to address each of the comments to improve our paper.  Here, we will answer in red between lines for readability purposes and in red in the manuscript itself.

The manuscript by Iván J. Núñez-Gil et al. entitled “Post-COVID 19 symptoms, vaccination and heart disease. Results from HOPE 2” aimed to characterize the outcomes and the impact of underlying heart diseases after surviving a COVID19 hospitalization.

I read with great interest this paper. The abstract summarizes the general significance of the manuscript; however, the title is not catchy and it does not sum up the meaning of the article.

Maybe. We modified it, following also the reviewer 2 comments.

Moreover, some major issues need to be addressed to improve the significance of the manuscript:

- In figure 3 the image of man and woman are not suitable to indicate the presence or absence of heart disease. Please, replace them.

Ok, done. We produced a new figure.

- It would be interesting to know the main basal blood tests of the two populations (for example hemoglobin or creatinine)

Sure, we added it to table 1.

- Cardiovascular (CV) involvement is a crucial complication in COVID-19, and no strategies are available to prevent or specifically address CV events in COVID patients. The identification of molecular partners contributing to CV manifestations in COVID-19 patients is crucial for providing early biomarkers, prognostic predictors and new therapeutic targets. Specifically, miRNAs have been proposed as valuable biomarkers and predictors of both cardiac and vascular damage occurring in SARS-CoV-2 infection. Consequently, this article could be discussed: Izzo C, Visco V, Gambardella J, et al. Cardiovascular implications of miRNAs in COVID-19 [published online ahead of print, 2022 Jul 2]. J Pharmacol Exp Ther. 2022;JPET-MR-2022-001210. doi:10.1124/jpet.122.001210.

Thanks for the suggestion, it is a relevant paper. We added it in the discussion area.

-As reported in limitations, the authors are not able to provide the relative weigh of the different heart disease in the patient’s outcome. Accordingly, it would be important to understand whether in “heart disease(+)”patients  due to an arrhythmia, the arrhythmia was still ongoing at the time of enrollment or was in the past. Similarly, did “heart disease(+)” patients due to valve disease have moderate/severe or only mild valve disease?

Absolutely, this is very important point. We comment on that in the methods and the limitations area. We accepted as heart disease of any kind when the patient had that diagnosis and was receiving specific treatment or follow up. Thus, for instance, a mild MR without treatment was not considered relevant. This was decided in a case per case basis, by the local research team.

- Some numbers of the bibliography are reported in the text without square brackets. Please check them.

Correct, we reported bibliography with superscript number. We change them all.

- Line 330: there is a typo “About Regarding”.

Thank you so much. Corrected.

Round 2

Reviewer 2 Report

Thank you for your changes; however, the titles of the tables should be written above the table rather than below it; additionally, reference (2), you cited it extensively in the introduction and discussion sections, which may lead to plagiarism throughout the manuscript; please, include additional references with it.

Author Response

We thank you for taking the time to review our manuscript in this second review. Here, again, we will answer in red between lines for readability purposes and in red in the manuscript.

Reviewer 2

Thank you for your changes; however, the titles of the tables should be written above the table rather than below it; additionally, reference (2), you cited it extensively in the introduction and discussion sections, which may lead to plagiarism throughout the manuscript; please, include additional references with it.

Thank you so much for your help. We corrected the issue with the tables. Following the same principle, we reordered the titles of the figures, also. The second reference is a metanalysis. We based our design in this paper (the idea). Thus, we feel it is fair to recognize that. Anyway, we modify some phrases and cite more widely the references in order to avoid the problem you mention.

Reviewer 3 Report

Thanks to the authors for the changes made; however, there is an error with the bibliography: 22 bibliographic sources, but only 21 inserted in the text. Probably, the miRNA article is number 22 and not 21.

Author Response

We thank you for taking the time to review our manuscript in this second review.  We will answer here and in red in the manuscript.

We appreciate your comment.  It helps a lot. Thank you so much again. We correct it.